# An Analysis of Factors Shaping Vaccine Attitudes and Behaviours in a Low-Trust Society Based on Structural Equation Modelling—The Case of Poland’s Vaccination Programme against COVID-19

**DOI:** 10.3390/ijerph192214655

**Published:** 2022-11-08

**Authors:** Michał Wróblewski, Andrzej Meler, Joanna Stankowska, Ewa Kawiak-Jawor

**Affiliations:** 1Institute of Sociology, Nicolaus Copernicus University, 87-100 Toruń, Poland; 2The Łukasiewicz Research Network Institute of Organisation and Management in Industry, 00-879 Warszawa, Poland

**Keywords:** COVID-19, vaccination programme, low-trust society, political polarisation

## Abstract

This study focuses on factors that shape vaccine attitudes and behaviours in the context of a low-trust society. Our analysis focuses on the Polish vaccination programme against COVID-19, primarily on (1) the evaluation of the information campaign, (2) trust in the institutions, (3) trust in other people, (4) attitudes toward vaccine safety and efficacy, (5) attitudes toward restrictions related to vaccination (e.g., restricted access to certain services for unvaccinated persons) and the introduction of mandatory vaccination, (6) the evaluation of the government’s actions during the pandemic, and (7) political preferences. The study was conducted with a sample of 1143 adult residents in Poland (CATI). The explanation of the factors determining the COVID-19 vaccine was based on structural equation modelling (SEM). The model showed that the declared fact of vaccination was largely determined by a positive attitude toward restrictions related to vaccination and trust in vaccines. The formation of the provaccine attitude was to an extent determined by the assessment of the government’s campaign and actions during pandemic. While institutional trust had a positive effect on support for the ruling coalition (0.56), the latter on its own had the opposite effect (−0.61) on the formation of provaccine attitude. In the group who both trust institutions and support the parties currently in power, there are more of those who simultaneously reject the restrictions and mandatory vaccination and remain sceptical about the safety and efficacy of COVID-19 vaccines than those who both trust in the vaccine safety and efficacy and accept the restrictions and mandatory vaccination. This indicates that in the context of strong political polarisation, ideological affiliations may play a greater role in shaping vaccine attitudes and behaviours than institutional trust.

## 1. Introduction

The COVID-19 pandemic has undoubtedly posed a major challenge for health care systems worldwide [1,2]. In addition to the management of the limited hospital infrastructure and resources, another pressing issue was to plan and effectively implement the vaccination programmes. The vaccination of the largest percentage of the population in the shortest time possible was challenging not only organisationally and logistically [3] but also because it was necessary to convince people to accept a vaccine. Research conducted before the COVID-19 pandemic shows that factors such as social trust, knowledge, risk perception and experience with the health care system play a key role in shaping vaccine attitudes and behaviours [4].

The recent health care crisis prompts questions about the course of the pandemic and how it may contribute to the success or failure of vaccination programmes, as well as about the impact of the local context in which such programmes are implemented. Particularly interesting is the context of low-trust societies, an area less explored by researchers, especially in terms of the COVID-19 pandemic. Studies from Hong Kong indicate that such societies may present different models of compliance with restrictions [5,6]. This raises the following questions: To what extent are vaccine attitudes shaped by trust in low-trust societies? How much do they depend on other factors, for example, political preferences and demographic variables (e.g., age)? Another critical issue is the motivation related to vaccines: How is it formed in such societies and what is the persuasive potential of vaccination promotion campaigns?

This study focuses on Poland as an example of a low-trust society [7,8]. In Poland, public trust in the government has been steadily decreasing in recent years; compared with other EU member states, it was the lowest even before the pandemic [9]. Similarly, the level of trust in medical expertise and the health care system is very low in Poland, as confirmed by studies conducted both before [10,11] and during the pandemic [12]. Research also shows low social capital levels, primarily in terms of trust and distrust in relationships with other people [13,14].

This article presents an analysis of factors shaping vaccine attitudes and behaviours based on an empirical study conducted with a representative sample of the Polish population. Our analysis focuses primarily on (1) the evaluation of the information campaign, (2) trust in the institutions, (3) trust in other people, (4) attitudes toward vaccine safety and efficacy, (5) attitudes toward restrictions related to vaccination (e.g., restricted access to certain services for unvaccinated persons) and the introduction of mandatory vaccination, (6) the evaluation of the government’s actions during the pandemic and (7) political preferences. Statistical data were analysed using structural equation modelling (SEM) to indicate interactions between the tested factors and estimate the effects of various factors on the formation of vaccine attitudes.

### 1.1. Factors Shaping Vaccine Attitudes and Behaviours

That vaccine, as a crucial area of public health, has long continued to raise multiple social controversies. Ample information has been collected regarding the factors that shape vaccine attitudes and behaviours, which can be divided into two groups. The first includes actions taken by authorities and broadly defined public institutions to convince a given population to receive the vaccine. The second comprises social factors such as demographic variables and social or institutional trust, which can have a negative or positive effect on people’s attitudes toward vaccination.

Three categories can be identified concerning activities undertaken to shape vaccine attitudes and behaviours: (1) increasing the demand for vaccination in the population (e.g., educational and information campaigns, vaccination as a condition to accessing educational institutions), (2) ensuring wide access to vaccination (e.g., vaccination programmes extended to other than medical areas such as schools and workplaces, complete reimbursement of vaccination costs) and (3) provider-based interventions by entities directly involved in vaccinations (e.g., phone calls from medical facilities to remind patients about the upcoming vaccination date) [15].

Given the subject of this article, particular attention should be paid to interventions in the form of communication with patients. Communication requires specific narrative strategies if it is to be successful in persuading people to receive a vaccine [16,17,18]. Emphasis is also placed on transparency regarding the vaccine risks and safety [19]. Effective communication must consider the attitudes of different target groups [20] and various beliefs about vaccines [21].

As no intervention intended to promote vaccination can operate in a vacuum, social factors shaping vaccine attitudes and behaviours are equally important. Many studies highlight the role of various types of trust [22,23], in particular, in the medical system [24] and the government [25]. Vaccines are part of the expert system [26,27], whose social acceptance requires trust due to barriers related to expert knowledge.

Trust can not only determine vaccine attitudes and behaviours but also contribute to the effectiveness of actions implemented by public authorities and institutions for shaping these attitudes and behaviours. Trust in health care professionals can decide how information about vaccination is obtained: the lower the trust levels, the stronger the tendency to address the informal sources of knowledge and thus succumb to vaccine scepticism [28]. Lower levels of trust in the government may also reduce the effectiveness of public campaigns promoting vaccination [29]. The level of trust in the health care system may translate into increased effectiveness in the introduction of mandatory vaccination [30]. In contrast, distrust of the government may result in negative attitudes toward vaccination both among adults [31] and parents in the case of childhood immunisations [25].

In the discussions of trust and vaccine attitudes, studies pointing to a relationship between social capital and attitudes toward vaccination seem inconclusive. On the one hand, social capital can emerge as a resource supporting individuals in their choice of health practices, at the level of both emotional support and the provision of institutional resources (e.g., in the form of information or access to specific health services) [32,33]. On the other hand, some studies reveal the negative impact of bonding social capital (in homogeneous groups such as families) on the acceptance of childhood immunisations [34,35]. In these studies, the respondents declaring stronger family relationships were more likely to present a negative attitude toward vaccination.

Research shows that vaccine attitudes and behaviours can be influenced by many factors. While trust is frequently suggested as an explanatory variable, it is vital to ask what other factors it is related to when shaping attitudes toward vaccination and whether it is the dominating factor in all contexts.

The COVID-19 pandemic offers some insight into this problem. The pandemic situation seems to have confirmed, prima facie, many previous observations about vaccine attitudes and behaviours. For example, before the launch of the COVID-19 vaccination programmes, it was implied that people with high social capital, measured as trust in other people and health care institutions, would make more rational decisions regarding their health and vaccination [36]. Studies conducted after the vaccines became available appear to confirm this position. People with low levels of generalised trust tend to be more sceptical and vaccine-hesitant [37]. Research also confirms a relationship between distrust in the government and vaccine scepticism [38,39,40]. Negative experiences with the health care system may also translate into lower satisfaction with medical services and thus affect vaccine hesitancy [41].

However, the context of COVID-19 is largely unique, as shown by research, which problematises some previously well-established claims about the factors shaping vaccine behaviours and attitudes. In terms of communication strategies, transparency emerges as a good example. While the communication of negative information about vaccines can increase public trust in the health care system, it may also reduce the acceptance of vaccines in the short term [42]. Another important context that distinguishes the COVID-19 pandemic is its unpredictability. Given the randomness of the SARS-CoV-2 virus mutations, the efficacy of vaccines against COVID-19 may decrease, affecting the methods of reporting about the effectiveness of vaccination programmes. A strategy based exclusively on the communication of vaccine benefits (e.g., vaccines as means to return to regular life), with no reference to restrictions, can have a diametrically opposite effect to what is intended, for example, in the form of a loss of public trust in vaccination as a reasonable course of action [43].

Certain challenges were indeed faced by vaccination programmes before and have only been exacerbated by the COVID-19 pandemic. The impact of conspiracy theories, which are as old as vaccination itself, has increased significantly due to both the geopolitical context (Russian manipulations in the global discourse on vaccines [44,45]) and the rapid growth of social media [46].

Age has emerged as a specific factor in the COVID-19 pandemic and in shaping vaccine attitudes and behaviours. Given the specific character of COVID-19, the elderly have been recognised as a high-risk group. Consequently, many activities and communications related to preventive measures (e.g., shopping hours for senior citizens, food delivery, easier access to health services) were addressed primarily to people over 60 [47]. This kind of discursive framing of COVID-19 may have had an impact on vaccine attitudes. Studies show that age is an important predictor of vaccine attitudes in the case of COVID-19. Younger people are more likely to be sceptical about vaccination [48] and delay their decision to receive a vaccine until the last minute [49]. The key role of age is also indicated by meta-analyses of available studies [50].

The factors shaping vaccine attitudes and behaviours can be determined not only by the unique pandemic aspects but also by the specific context including, for example, political beliefs. While certain studies indicate the influence of the latter on vaccine attitudes and behaviours [51], others reveal no such relationship [52]. Consequently, the question arises about the extent to which a particular political context, characteristic of the given time and place, can affect people’s attitudes towards vaccination. Another important contextual factor may be a country’s level of development. There is greater public acceptance of COVID-19 vaccination in LMICs than in high-income countries due to the presence of greater infectious disease risks in developing countries [53].

### 1.2. The COVID-19 Vaccination Programme in Poland

Prepandemic research shows a relatively high social acceptance of vaccination in Polish society, with young adults emerging as the most sceptical group [54]. However, public trust in vaccine safety decreased in Poland compared with other EU countries between 2015 and 2017, as did the measles vaccination coverage [23]. Perhaps this trend influenced the perception of COVID-19 vaccines, which was not positive even before the launch of the vaccination programme in December 2020. In November 2020, only 36% of Poles were willing to receive a COVID-19 vaccine, and 47% had no such intention [55]. In an international comparative study of 33 countries from December 2020, Poland had one of the lowest rates of willingness to receive a COVID-19 vaccine [56].

The National COVID-19 Vaccination Programme was launched on 27 December 2020 and was initially addressed only to health care professionals. Over time, it became accessible to seniors and selected professional groups (e.g., teachers and representatives of uniformed services). The schedule for the inclusion of the respective social groups and age categories in the programme was altered several times, and the changes were not always communicated transparently and comprehensibly. The registration for the COVID-19 vaccination for all adult citizens started in May 2021. From the beginning, the National Vaccination Programme was promoted in an information campaign organised by the Polish government (specifically, the Chancellery of the Prime Minister). The campaign had three priorities: (1) to convince people to receive a vaccine (e.g., emphasis on safety and efficacy); (2) to provide information (on registration options and vaccination points) and (3) to fight disinformation on social media.

In mid-July 2021, Poland’s vaccination programme covered 42.09% of the population (compared with an average of 42.02% in the EU) and ranked 5th among the 10 most populated EU member states. Currently (as of 6 June 2022), the vaccination coverage in Poland is less than 60% (compared with an average of over 73% in the EU) and is in the penultimate place, which raises questions about the reasons for this state of affairs. Despite the problems in increasing the vaccination coverage, the Polish authorities have decided not to follow the example of certain EU countries to implement stronger measures, e.g., COVID-19 passes and mandatory vaccinations for selected groups or the entire adult population. Such solutions were implemented in Germany, France, Austria, Italy and Spain. Analyses comparing the situation between Poland and Lithuania, which has introduced a COVID-19 certificate policy, show the positive impact of this solution [57,58].

The very idea of introducing these measures caused tensions in the current ruling coalition of several conservative parties. Representatives of the largest political force in the coalition (Law and Justice; in Polish: Prawo i Sprawiedliwość, PiS), to which the Minister of Health belongs, tried to make vaccination obligatory for selected professional groups and authorise employers to verify employee compliance with this obligation. However, this met with strong opposition from some representatives of the main coalition party and the second political force (Solidary Poland; in Polish: Solidarna Polska, SP), which threatened to leave the coalition. Given the problem with parliamentary majority, their opposition proved effective. Eventually, no regulations were implemented regarding the partially compulsory vaccinations or any disclosures of information on employee vaccination status.

Studies conducted so far point to a number of potential factors shaping vaccine attitudes in Poland as also related to political preferences. Walkowiak and Walkowiak [59] link vaccine scepticism in the context of COVID-19 primarily to people who do not participate in elections and express distrust of public institutions.

## 2. Methods

The study was conducted between 30 December 2021 and 16 January 2022 with a sample of 1143 adult residents in Poland. The sample was representative in terms of age, gender and place of residence. Data were collected using computer-assisted telephone interviews (CATI).

The explanation of the factors determining the COVID-19 vaccine was based on structural equation modelling (SEM). This method has already been employed in studies on vaccine attitudes [60,61]. The latent variable Y was defined as the reception of a vaccine by a respondent, which was verified with a dedicated question in the questionnaire (‘Are you fully vaccinated against COVID-19?’).

Based on the previously cited studies on factors influencing vaccine behaviour, we created a conceptual model. In addition to the factors cited in the studies (trust, age, political views), we also included factors related to the management of the pandemic and the evaluation of the promotional campaign, since COVID-19 vaccination took place in the form of a programme implemented by the authorities. In doing so, we made two assumptions: (1) vaccine behaviour will be influenced by a certain vaccine attitude understood as a set of beliefs about vaccines and the vaccination programme; (2) the influence of certain factors on vaccine attitude will be mediated by other factors. At the same time, we assumed that political factors, at both the level of political views and the evaluation of government actions and communication, would mediate the influence of factors such as trust and age on vaccine attitudes.

The following constructs were used in the model: attitudes toward vaccination and related restrictions (M1), attitudes toward the information campaign and the government actions related to the pandemic (M2), institutional trust (government, police, courts of law, local authorities) (X1) and social trust (neighbours, friends, people met for the first time, followers of other denominations, representatives of other countries) (X3). In addition, our model included two more variables, age (X2) and attitudes toward the government coalition (M3). The latter (M3) was composed of the aggregated responses to the question about political preferences (‘Which party would you vote for if parliamentary elections were held today?). Respondents’ answers were aggregated into two groups: (1) indication of the parties from the ruling coalition and (2) indication of the opposition parties or no indication of any party. All variables building the respective constructs are presented in Table 1.

The attitude toward the information campaign was measured using two types of questions. The first focused on the overall assessment of the campaign (‘What is your general opinion about the campaign promoting the COVID-19 vaccination?’) and the second on the assessment of the quality of the information provided in the campaign (‘What is your opinion about the quality of the information provided in the campaign promoting the COVID-19 vaccination?’). In this question, respondents were asked to score the statements using a four-point Likert scale from strongly agree to strongly disagree. Two statements were used: ‘I know how to register for the vaccine’ and ‘I know where I can get vaccinated’. The questions measuring the level of trust in institutions and people were modelled on the World Values Survey Social Capital Scale [62,63].

The M1, M2, X1 and X3 constructs were tested for test score reliability. Cronbach’s alpha (>0.6) was considered satisfactory, enabling us to conclude that the variables within the constructs measured the same phenomena [64].

Given that the observable variables were ordinal scales, the diagonally weighted least squares estimator, recommended for this type of data, was used for model calculation [65]. The number of observations employed in the model after subtracting those with missing data values was 799. The comparative fit index (CFI) was 0.946 and the Tucker–Lewis Index (TLI) 0.938, while the RMSEA equalled 0.075, indicating good fit of the model [66,67]. The model was calculated in the R environment using lavaan.

## 3. Results

The model explains the reception of vaccination in 59% (Y), the formation of positive attitudes toward vaccines and restrictions (M1) in 40%, and the assessment of the vaccination campaign and government actions in 57% (M2).

The model (see Figure 1) showed that the declared fact of vaccination was largely determined by a positive attitude toward restrictions related to vaccination and trust in vaccines (M1): a standardised index indicated that an increase in M1 by one standard deviation raised the vaccination chance by 0.7 standard deviation in Y [68] (p. 349). At the same time, the formation of attitude M1 was to an extent determined by the assessment of the government campaign and actions (M2), with each increase in this assessment of one standard deviation resulting in a rise in the M1 attitude by 0.92 standard deviation.

The statistical model is shown in Figure 2. The assessment of the government campaign and actions in our model was mainly influenced by two factors: age and institutional trust, whose standardised effects were 0.37 and 0.66, respectively (see Figure 1, Table 2). The effect of public trust was much smaller but also statistically significant. Interestingly, the higher the respondents’ trust in other people, the lower their assessment of the government campaign and actions (standardised effect = −0.19).

While institutional trust had a positive effect on support for the ruling coalition (0.56), the latter on its own had the opposite effect (−0.61) on the formation of positive attitudes toward vaccines (see Figure 1, Table 2). In other words, in the population of people who both trust institutions and support the parties currently in power, there are more of those who simultaneously reject the restrictions and remain sceptical about the safety and efficacy of COVID-19 vaccines than those who both trust in the vaccine safety and efficacy and accept the restrictions. Given the positive covariance between M2 and M3 (0.73), it can be concluded that the government campaign and actions divided the supporters (as well as opponents) of the coalition differently, affecting their attitudes toward vaccines and restrictions.

None of the three independent variables, X1, X2 or X3, proved to have a direct effect on Y (*p*-value for regressions Y~X1, Y~X2 and Y~X3 greater than 0.05); however, they had a direct effect on M1, specifically attitudes conducive to the reception of a vaccine. It is worth noting that institutional trust (X1) had an inverse direct effect on M1 (attitudes toward vaccine safety and efficacy and toward restrictions): the greater the trust in institutions, the lower the support for M1. Nevertheless, the effect mediated through M2 (the assessment of the government campaign and actions related to the pandemic) not only reversed this impact (higher trust levels in institutions were conducive to the M1 attitude) but also strengthened it (see Table 3). The inverse impact was observed when the political factor (support for the ruling party, M3) was considered as a mediator.

## 4. Discussion

Unsurprisingly, a positive attitude toward the safety and efficacy of vaccination affected vaccine behaviours. Interestingly though, in our model, this attitude was closely linked to the acceptance of mandatory vaccinations, restrictions for the unvaccinated and an employer’s access to employee vaccination status. This means that the term ‘vaccine attitudes’ covers a package of views not only regarding the vaccines themselves but also on the functioning of the entire vaccination programme. It can be assumed that vaccinated people are highly aware of the COVID-19 risks and are therefore also convinced that given the low vaccination coverage (as in Poland), the introduction of mandatory vaccinations or restrictions for the unvaccinated is a necessary solution. For these people, vaccines are a benefit not only in terms of individual health but also socially; for example, they contribute to safety in the public space, which depends on the decisions of others. The positive attitude toward restrictions for the unvaccinated, employer’s access to employee vaccination status and mandatory vaccinations indirectly expresses the expectation that others, even if not of their own volition, should behave responsibly for the sake of public safety and should be vaccinated.

In our analysis, the assessment of the vaccination campaign and the government actions during the pandemic proved to be interrelated, similar to the attitudes toward vaccine safety and efficacy and restrictions related to vaccination. In other words, those of our respondents who had a positive opinion about the campaign were also more likely to positively assess the government’s actions during the pandemic. This suggests that the opinion of the campaign addressees can be influenced not only by the quality of its content but also by the assessment of the government’s decisions taken during the entire pandemic. This may be due to the respondents tending to identify the campaign, and perhaps also other organisational aspects of the vaccination programme, with the government (which was indeed responsible for both). This strong link between the assessment of the campaign and the assessment of the government’s performance is indicative of the significant role of the latter in the context of a health crisis. It should be remembered that in our model, a positive assessment of the government and the campaign proved to be associated with a positive attitude toward vaccination and restrictions, which in turn has a strong effect on vaccine behaviours. This means that in the context of an epidemic, the quality and effectiveness of government crisis management emerge as important resources in shaping rational health behaviours, including during a vaccination programme. A relationship between the assessment of government actions and the attitude to COVID-19 vaccination has also been found in other studies [69,70].

In the context of a low-trust society, the relationships between the studied phenomena and trust are particularly interesting. On the one hand, a positive, albeit weak, correlation (0.25) was found between trust in other people (X3) and attitudes toward vaccine safety and efficacy and related restrictions (M1). On the other hand, the correlation between X3 and the assessment of the government campaign and actions (M2) was negative (−0.19). Considering the indirect effects, the following relationship was found: the respondents who trusted other people but had a negative opinion about the government campaign and actions tended to be sceptical of the vaccine safety and efficacy and related restrictions (although this effect was weak: −0.026).

How do we explain this? Assuming that the level of trust in other people may be related to the size of personal social networks, we can presume that those who trust other people enter into more diverse interactions than those who are distrustful. It is said that social network types can have an impact on health behaviours, although this impact may not always be uniform [71]. People with wide social contacts have access to health information from various and not necessarily official sources: ‘being socially connected has the potential for increasing the odds of being exposed to new alternative treatment types found to be efficient in certain medical conditions’ [71] (p. 5). This observation is confirmed in certain studies, for example, on people functioning as bridges between different social networks who are more likely to adopt complementary and alternative medicine (CAM) [72]. While a positive attitude toward CAM does not directly translate into vaccine hesitancy [73,74,75], access to diverse information sources, including those from alternative knowledge systems, has the potential to reduce the effectiveness of information campaigns. Consequently, turning to social networks may be an alternative to official sources as a strategy for shaping vaccine behaviours.

Another aspect is the relationships between institutional trust (X1) and attitudes toward vaccine safety and efficacy with related restrictions (M1). Given the extant research on the impact of trust on vaccine attitudes, our model surprisingly revealed a negative correlation between these two phenomena. However, this relationship looked slightly different when a mediating factor was included, attitudes toward the ruling coalition (M3) or the assessment of the government campaign and actions related to the pandemic (M2). Two trajectories could be observed here in terms of the relationship between trust in vaccination and attitudes toward vaccines. On the one hand, people who trusted institutions and supported the ruling coalition tended to be critical of vaccine safety and efficacy and related restrictions. On the other hand, people who trusted institutions and had a positive opinion about the government campaign and actions had a more positive attitude toward vaccines and restrictions. As our model shows, the latter largely determines vaccine behaviours.

The results of our study indicate that institutional trust has an impact on vaccine attitudes and behaviours, depending on the circumstances. Elevated levels of trust in the government, courts of law, police and local authorities may translate into a positive attitude toward vaccination and the acceptance of related restrictions, but not in the case of those who support parties of the ruling coalition. Being for the ruling coalition is a factor that mediates the relationship between institutional trust and vaccine attitudes so strongly that it can change its vector.

This phenomenon can largely be explained by the context of Polish politics and the specificity of the debates held during the implementation of the vaccination programme. As mentioned, the idea of mandatory vaccinations for selected professional groups and for the verification of employee vaccination status by employers caused tensions within the government coalition. Moreover, although the national vaccination programme was a government initiative implemented by the Chancellery of the Prime Minister, some politicians associated with the ruling coalition publicly expressed their scepticism about vaccine safety and efficacy during the programme (The most well-known case was that of Barbara Nowak, Chief Education Officer in the Małopolskie Voivodeship, appointed to this position by the main party in the government coalition (Law and Justice), who in an interview called the COVID-19 vaccines ‘a medical experiment with unknown consequences’ (https://www.rp.pl/edukacja/art19262451-barbara-nowak-o-szczepionkach-nie-sa-do-konca-znane-konsekwencje-tego-eksperymentu, accessed on 1 October 2022). This statement was made in the context of mandatory vaccinations for teachers. Despite the criticism and pressure from the Ministry of Health, Barbara Nowak was not dismissed from her position.). Our results suggest that both the political conflict and the voices of certain politicians may have contributed to a negative attitude toward vaccine safety and efficacy and related restrictions, as well as mandatory vaccinations, which as our model shows might have translated into vaccine behaviours.

In the analysis of their impact on vaccine behaviours and attitudes, political preferences were considered through their binary distribution (supporting the ruling coalition vs. not supporting the ruling coalition). This approach refers to partisan polarisation defined as a tendency to perceive the ‘political life of a community (…) as a battle between two partisan sides’ [76] (p. 2). Electoral preferences alone (voting for one party or the other) do not necessarily translate into strong partisan polarisation; however, the fact that certain phenomena (e.g., related to the pandemic) are perceived through the prism of ideological affiliations may be indicative of this form of attitude divergence. Given the context of the political debate on vaccination in Poland, one could assume that this topic has strongly polarised Polish society. Our results prove this to be the case. It is worth highlighting that partisan polarisation existed in Poland before the COVID-19 pandemic, as confirmed by the results of the European Social Survey (2002–2020), where Poland was described as a highly polarised country along with Spain, Greece, Turkey, Croatia and Hungary [76].

In broader terms, political polarisation generally leads to a fierce public debate, including in the case of topics related to scientific expertise. Ideological divisions between the supporters and opponents of certain political camps are increasingly evident also in the attitude to science. The example of the United States shows that partisan polarisation can be strongly linked to science scepticism, while attitudes towards phenomena such as climate change can be affected by political views [77]. Strong political polarisation during the pandemic resulted in the replacement of the experts’ debates on the safety, risks and effectiveness of health policies with political discourse and ideological preferences [78]. While enhancing the differences between the élites (establishment) and the rest of society, populist discourse is also conducive to the dissemination of conspiracy theories [79], which may have a negative impact on vaccine attitudes and behaviours.

In our model, attitudes toward the government coalition were a mediating factor in the relationship between institutional trust and attitudes toward vaccines and related restrictions. This indicates that in the context of strong political polarisation, ideological affiliations may play a greater role in shaping vaccine attitudes and behaviours than institutional trust. This prompts a question about a link between partisan polarisation and attitudes to vaccination in low-trust societies. While further research is necessary, such a link has been confirmed in some studies. For example, attitudes toward COVID-19 vaccination in Brazil, a country of strong internal conflicts, have been associated with political polarisation [80]. Moreover, Brazil is also a country where trust in the government is low [9]. A link between partisan polarisation and attitudes toward COVID-19 vaccination has also been found in the United States [81], where trust in the government has been historically low for more than 60 years.

Unlike other EU countries, Poland has not decided to introduce COVID-19 passes or make vaccination compulsory. Nevertheless, our model also covered attitudes toward mandatory vaccinations and restrictions for the unvaccinated (M1). The relationship between support for the government coalition and M1 attitude suggests that Poland’s decision not to follow in the footsteps of Germany, Italy, Austria or Spain was dictated by political factors. The fact that respondents declaring support for the ruling parties were at the same time critical of the idea of mandatory vaccinations could indicate a significant political risk. The vaccination obligation might provoke a part of the government electorate to oppose and join the party that openly supports anti-vaccine movements (Confederation, in Polish: *Konfederacja*, is represented in the Polish parliament). Given the previous experience with compulsory vaccinations in other countries, this risk must be recognised as real. Italy is an example where the list of compulsory vaccinations was extended given the new outbreaks of measles. While it increased the vaccination coverage in the population, this decision also led to social controversy, giving popularity to political forces opposed to vaccination [82].

## 5. Conclusions

Despite the repetitiveness of certain elements, each pandemic seems to be a unique phenomenon following its own specific scenario. Political and social contexts can have a decisive impact on how a pandemic will affect specific populations and societies. The same applies to vaccine behaviours and attitudes. While abundant prepandemic research reveals many regularities in this area, the context of both COVID-19 and individual societies can shed new light on our current knowledge.

The study presented in this article highlights the importance of several issues that undoubtedly call for further research. First is the role of trust in shaping vaccine attitudes and behaviours. Although numerous studies confirm the positive impact of trust in institutions, government and the health care system on vaccination acceptance levels, our model indicates that in the conditions of the pandemic and a low-trust society, this relationship is not so clear. On the one hand, low-trust societies lack the crucial resources for making rational health decisions, but on the other hand, it is possible to build vaccination acceptance in such contexts despite the generally low level of institutional trust. In our model, the relationship between the assessment of the government campaign and actions and attitudes toward vaccine safety and efficacy and related restrictions reveals a positive potential that lies in decisions made for health crisis management and the implementation of the vaccination programme. The second issue is connected with political polarisation. This phenomenon too has a negative impact on public health. Poland’s example appears to illustrate that the lack of political consensus regarding vaccination and a heated public debate on this topic can have a negative effect on vaccine attitudes and behaviours.

Based on our research, the following recommendations can be made: (1) special attention should be paid to the shaping of communication by poly-tic actors; their role in shaping pro-health attitudes is particularly important in the context of a politically polarized society; (2) treatments aimed at shaping vaccine attitudes should refer to actors who enjoy special social authority in a low-trust society, e.g., local organizations and opinion leaders.

## Figures and Tables

**Figure 1 ijerph-19-14655-f001:**
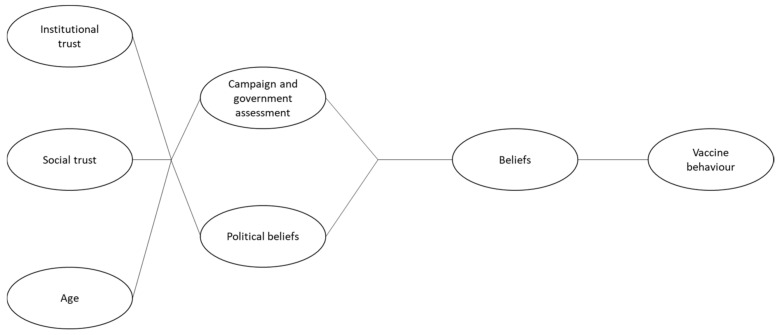
Conceptual model.

**Figure 2 ijerph-19-14655-f002:**
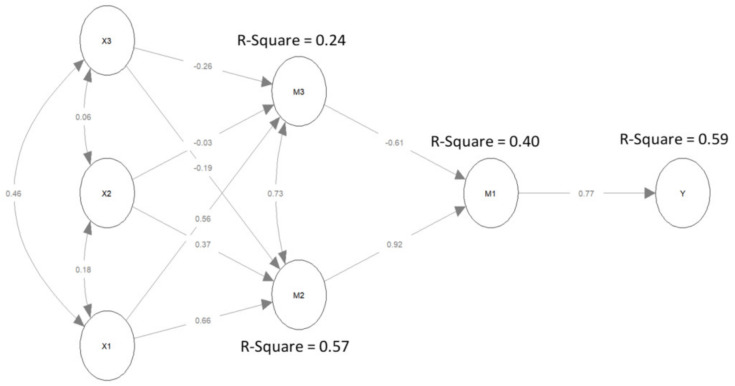
SEM model.

**Table 1 ijerph-19-14655-t001:** Constructs.

Latent	Op	Observed	Coeff.	S.E.	z	*p*-Value	Std. Coeff.
Y	=~	vaccinated	1.00	0.00			1.00
M1: Attitude toward restrictions and vaccines Cronbach’s alpha = 0.90	=~	**RE1** Restriction Evaluation: Should there be restrictions for unvaccinated people in public places?	1.00	0.00			0.83
	=~	**RE2** Restriction Evaluation: COVID-19 vaccination should be mandatory for all adults.	0.99	0.03	38.75	0.00	0.84
	=~	**RE3** Restriction Evaluation: Employers should have access to information on which employees are vaccinated against COVID-19.	0.98	0.03	38.11	0.00	0.82
	=~	**VA1** Vaccine Attitude: COVID-19 vaccines are effective in protecting people from getting infected.	0.89	0.02	37.96	0.00	0.85
	=~	**VA2** Vaccine Attitude: COVID-19 vaccine research is reliable and can be trusted.	0.86	0.02	37.97	0.00	0.85
	=~	**VA3** Vaccine Attitude: There is no risk of severe complications from COVID-19 vaccines.	0.58	0.02	30.73	0.00	0.60
M2: Campaign and government assessment Cronbach’s alpha = 0.64	=~	**CA1** Campaign Assessment: What is your general opinion about the campaign promoting COVID-19 vaccination?	1.00	0.00			0.69
	=~	**CA2** Campaign Assessment: I know how to register for the vaccination.	0.31	0.02	13.25	0.00	0.29
	=~	**CA3** Campaign Assessment: I know where I can get vaccinated.	0.23	0.02	11.67	0.00	0.25
	=~	**CA4** Government Assessment: What is your opinion about the government actions to fight the coronavirus epidemic in Poland?	1.03	0.05	21.33	0.00	0.66
M3: Political factor	=~	**POL**: Attitude towardthe ruling coalition	1.00	0.00			1.00
X1: Institutional trust Cronbach’s alpha = 0.74	=~	**TR1** Trust in the government	1.00	0.00			0.71
	=~	**TR2** Trust in the police	0.99	0.05	20.37	0.00	0.72
	=~	**TR3** Trust in the courts of law	0.72	0.04	18.57	0.00	0.56
	=~	**TR4** Trust in the local authorities	0.88	0.04	20.41	0.00	0.70
X2: Age	=~	Age	1.00	0.00			1.00
X3: Social trust Cronbach’s alpha = 0.73	=~	**PE1** Trust in neighbours	1.00	0.00			0.69
	=~	**PE2** Trust in people I know personally	0.75	0.06	13.56	0.00	0.63
	=~	**PE3** Trust in people I met for the first time	0.48	0.04	11.79	0.00	0.42
	=~	**PE4** Trust in people of different denomination	0.86	0.06	13.40	0.00	0.64
	=~	**PE5** Trust in foreigners	0.73	0.06	12.91	0.00	0.55

=~—**indicator**, used for latent variable to observed indicator in factor analysis measurement models.

**Table 2 ijerph-19-14655-t002:** Regressions and covariances.

Latent Variable	Op	Latent Variable	Coeff.	S.E.	z	*p*-Value	Std. Coeff.
Y	~	M1	0.27	0.01	37.46	0.00	0.77
Y	~	X1	−0.05	0.13	−0.39	0.70	−0.09
Y	~	X2	−0.01	0.02	−0.42	0.67	−0.02
Y	~	X3	−0.01	0.07	−0.14	0.89	−0.02
M1	~	X1	−0.58	0.11	−5.06	0.00	−0.36
M1	~	X2	0.44	0.04	10.73	0.00	0.34
M1	~	X3	0.44	0.04	10.73	0.00	0.25
M1	~	M2	1.44	0.13	11.54	0.00	0.92
M1	~	M3	−2.49	0.37	−6.76	0.00	−0.61
M2	~	X1	0.66	0.05	12.71	0.00	0.66
M2	~	X2	0.31	0.03	9.80	0.00	0.37
M2	~	X3	−0.21	0.05	−4.63	0.00	−0.19
M3	~	X1	0.22	0.02	10.80	0.00	0.56
M3	~	X2	−0.01	0.01	−0.75	0.45	−0.03
M3	~	X3	−0.11	0.02	−5.74	0.00	−0.26
M2	~~	M3	0.13	0.01	9.03	0.00	0.73
X1	~~	X2	0.17	0.03	6.87	0.00	0.18
X1	~~	X3	0.33	0.02	14.30	0.00	0.46
X2	~~	X3	0.05	0.02	2.25	0.02	0.06

=~—**predict**, used for regression of observed outcome to observed predictors. ~~—**covariance**.

**Table 3 ijerph-19-14655-t003:** Indirect effects.

Indirect Effect Path	Op	Coeff.	S.E.	z	*p*-Value	Std. Coeff.
X1_M2_M1	:=	1.02	0.13	7.81	0.00	0.63
X1_M3_M1	:=	−0.07	0.03	−2.78	0.01	−0.04
X2_M2_M1	:=	0.10	0.04	2.84	0.00	0.08
X2_M3_M1	:=	−0.02	0.01	−1.77	0.08	−0.01
X3_M2_M1	:=	−0.48	0.06	−7.71	0.00	−0.26
X3_M3_M1	:=	0.05	0.02	2.60	0.01	0.03
X1_M2_M1_Y	:=	0.29	0.04	7.05	0.00	0.51
X1_M3_M1_Y	:=	−0.02	0.01	−2.75	0.01	−0.03
X2_M2_M1_Y	:=	0.03	0.01	2.77	0.01	0.06
X2_M3_M1_Y	:=	0.00	0.00	−1.77	0.08	−0.01
X3_M2_M1_Y	:=	−0.13	0.02	−7.04	0.00	−0.21
X3_M3_M1_Y	:=	0.01	0.00	2.57	0.01	0.02

:=—operator defines new parameters which take on values that are an arbitrary function of the original model parameters.

## Data Availability

The dataset used during the current study are available under the following link: https://repod.icm.edu.pl/file.xhtml?fileId=12074&version=DRAFT (accessed on 1 October 2022).

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
