# Peer review of "An Analysis of Factors Shaping Vaccine Attitudes and Behaviours in a Low-Trust Society Based on Structural Equation Modelling—The Case of Poland’s Vaccination Programme against COVID-19"

_ijerph, 2022, doi:10.3390/ijerph192214655_

Round 1

Reviewer 1 Report

Ever since the first COVID-19 vaccine was authorized a heated debate over the usefulness of coronavirus vaccines emerged. This in turned resulted in the rise of antivaccine movement and high rates of vaccine hesitancy in various countries including Poland. Thus, the unquestionable strength of the manuscript is that its topic is very important and timely and the research itself was designed and described clearly. Moreover, while there are many research on the vaccine hesitancy in numerous countries including Poland still far too little of these research try to provide solid explanation on the social reasons of COVID-19 vaccine decision-making. This is of key importance because identifying factors associated with vaccine acceptance/hesitancy may inform effective vaccine messaging and campaigns. For all these reasons I believe that the reviewed manuscript fills the gap in the literature and may be of interest to the readers of the Journal.

At the same time, while I recommend this research for publication I have some minor comments/suggestions:

1. In the Introduction section a more detailed information about the discourse the pros and cons of the (Covid-19) vaccine could be described.

Additionally, it could be compared with the situation in other countries from the region. For example:

-- Machingaidze, S., Wiysonge, C.S. Understanding COVID-19 vaccine hesitancy. Nat Med 27, 1338–1339 (2021). https://doi.org/10.1038/s41591-021-01459-7.

-- Bass SB, Wilson-Genderson M, Garcia DT, Akinkugbe AA, Mosavel M. SARS-CoV-2 Vaccine Hesitancy in a Sample of US Adults: Role of Perceived Satisfaction With Health, Access to Healthcare, and Attention to COVID-19 News. Front Public Health. 2021 Apr 29;9:665724. https://doi.org/10.3389/fpubh.2021.665724.

-- Hudson A, Montelpare WJ. Predictors of Vaccine Hesitancy: Implications for COVID-19 Public Health Messaging. Int J Environ Res Public Health. 2021 Jul 29;18(15):8054. https://doi.org/10.3390/ijerph18158054.

2. Similarly, as the Authors describe the vaccination strategy of the Polish government it would be interesting to put it in a broader perspective. Did it differ from that in other countries from the region? For example, while describing the discussion on the (effectiveness of) policy regarding Covid-19 passes and mandatory vaccinations in Poland the Authors could refer to the existing research on the topic:

-- Walkowiak MP, Walkowiak D. Predictors of COVID-19 Vaccination Campaign Success: Lessons Learnt from the Pandemic So Far. A Case Study from Poland. Vaccines. 2021;9(10):1153. https://doi.org/10.3390/vaccines9101153.

-- Walkowiak MP, Walkowiak JB, Walkowiak D. COVID-19 Passport as a Factor Determining the Success of National Vaccination Campaigns: Does It Work? The Case of Lithuania vs. Poland. Vaccines. 2021;9(12):1498. doi: 10.3390/vaccines9121498. 

-- Walkowiak MP, Walkowiak JB, Walkowiak D. More Time, Carrot-and-Stick, or Piling Coffins? Estimating the Role of Factors Overcoming COVID-19 Vaccine Hesitancy in Poland and Lithuania in the Years 2021-2022. Vaccines. 2022;10(9):1523. https://doi.org/10.3390/vaccines10091523.

3. I am a bit surprised that apart from many demographics no information on such important factors as religion/spirituality and self-perceived status health was given. Meanwhile, research show that they also influence individuals’ attitudes towards vaccination. For example, research suggest that some communities and/or individuals were morally concerned over the fact that some vaccine manufacturers used abortion-derived fetal cell lines:

-- Zimmerman RK. Helping patients with ethical concerns about COVID-19 vaccines in light of fetal cell lines used in some COVID-19 vaccines. Vaccine. 2021;39(31):4242-4244. doi:10.1016/j.vaccine.2021.06.027

-- Garcia LL, Yap JFC. The role of religiosity in COVID-19 vaccine hesitancy. J Public Health (Oxf). 2021 Sep 22;43(3):e529-e530. doi: 10.1093/pubmed/fdab192.

Taking under consideration that Polish society, and especially its Eastern part and rural areas, where the vaccination rates was lower, are highly religious it could have played some role in Poland’s vaccination programme against COVID-19.

4. Finally, the paper would benefit from adding some recommendations suggesting possible guidelines that should be implemented in order to overcome the problem discussed in the manuscript.

Minor:

-- line 58: double reference: [13, 14] (Czapiński and Panek 2015; Marjański et al. 2019).

            Apart from these small remarks, I appreciate this research a lot and I wish to congratulate the Authors on their interesting paper and I recommend its publication. I am convinced that the issues raised in the article will help to understand some social factors that underline the success of social vaccination campaigns against COVID-19. It can also stimulate the discussion on the effective vaccine messaging by policymakers and healthcare professionals.

Author Response

Reviewer 1

Comment

Reaction

1

In the Introduction section a more detailed information about the discourse the pros and cons of the (Covid-19) vaccine could be described. Additionally, it could be compared with the situation in other countries from the region

Suggested literature references has been added.

2

Similarly, as the Authors describe the vaccination strategy of the Polish government it would be interesting to put it in a broader perspective. Did it differ from that in other countries from the region? For example, while describing the discussion on the (effectiveness of) policy regarding Covid-19 passes and mandatory vaccinations in Poland the Authors could refer to the existing research on the topic

Suggested literature references has been added.

3

I am a bit surprised that apart from many demographics no information on such important factors as religion/spirituality and self-perceived status health was given. Meanwhile, research show that they also influence individuals’ attitudes towards vaccination. For example, research suggest that some communities and/or individuals were morally concerned over the fact that some vaccine manufacturers used abortion-derived fetal cell lines

Religiosity was initially included in the conceptual model. However, in the computational model, the variable religiosity turned out to prevent statistical identification of the model, mainly due to its lack of linkage to the final dependent variable (vaccination) but also weak linkages to other variables. Therefore, following the recommendation that says that in case of identification problems, among other things, less significant variables and links should be eliminated from the model (Schumacker and Lomax 2015), the decision was made to remove this variable.

Schumacker, Randall E., i Richard G. Lomax. 2015. A Beginner’s Guide to Structural Equation Modeling. 4. wyd. Taylor & Francis.

4

Finally, the paper would benefit from adding some recommendations suggesting possible guidelines that should be implemented in order to overcome the problem discussed in the manuscript.

Recommendation has been added.

5

Line 58: double reference

The double reference has been removed.

Reviewer 2 Report

Journal: International Journal of Environmental Research and Public Health (MDPI)

Title: Analysis of factors shaping vaccine attitudes and behaviors in low-trust societies based on structural equation modeling. The cases of Poland’s vaccination program against COVID-19

    This manuscript explores the actors associated with vaccine attitudes and behaviors in the context of a low-trust society. Vaccine hesitancy is an important public health issue and more studies on this issue is needed. However, I have some concerns over the methods used in the study.

1.      Structural Equation Modelling (SEM) is a technique examining the consistency between the theory (model) and data. It is necessary to build a model first. Reviewing previous literature to come up with a justifiable model. The authors should explain why and they specified the relationships among variables in the model they adopted.

2.      Usually, a full SEM model contains 2 parts of a model, the measurement model (how you measure a latent variable/construct? For example, the questionnaire items used to measure attitude and items used to measure trust) and the structural model (relationships between latent variables/constructs).

3.      After fitting the model, the authors should report the Fit Index statistics (such as GFI, and CFI…). These statistics tell readers how well the data fit the model. If the level of consistency between the model and data is acceptable, then the model is not refuted, so we can continue to discuss specific relationships identified from the model. Sometimes, there is a need to revise the model in order to find a better fit and that would become the findings of the study.

4.      The writing is not clear. For example, in line 438-439, “our model indicates that in the conditions of the pandemic and low-trust society, this relationship is not so unambiguous.”  “not so unambiguous” is double negative and it’s confusing.

5.      In lines 448-450, “The example of Portugal shows that political consensus can be an important and positive factor in deciding the effectiveness of health crisis management and the success of the vaccination program.” This example of Portugal comes from nowhere. The authors never mentioned this example of Portugal in the text before.

Author Response

Reviewer 2

Comment

Reaction

1

Structural Equation Modelling (SEM) is a technique examining the consistency between the theory (model) and data. It is necessary to build a model first. Reviewing previous literature to come up with a justifiable model. The authors should explain why and they specified the relationships among variables in the model they adopted.

Conceptual model has been added (lines 224-238).

2

Usually, a full SEM model contains 2 parts of a model, the measurement model (how you measure a latent variable/construct? For example, the questionnaire items used to measure attitude and items used to measure trust) and the structural model (relationships between latent variables/constructs).

The measurement model is included in Table 1 (line 267). It contains a description of all constructs used in the SEM model with an indication of specific variables. We have not included it in the SEM model (figure 2) out of concern for visual clarity.

3

After fitting the model, the authors should report the Fit Index statistics (such as GFI, and CFI…). These statistics tell readers how well the data fit the model. If the level of consistency between the model and data is acceptable, then the model is not refuted, so we can continue to discuss specific relationships identified from the model. Sometimes, there is a need to revise the model in order to find a better fit and that would become the findings of the study.

We used the Fit Index statistics (CFI), Tucker-Lewis Index (TLI). Their indications are included in lines 269-274. Results show a good fit of the model.

4

The writing is not clear. For example, in line 438-439, “our model indicates that in the conditions of the pandemic and low-trust society, this relationship is not so unambiguous.”  “not so unambiguous” is double negative and it’s confusing.

This sentence has been changed.

5

In lines 448-450, “The example of Portugal shows that political consensus can be an important and positive factor in deciding the effectiveness of health crisis management and the success of the vaccination program.” This example of Portugal comes from nowhere. The authors never mentioned this example of Portugal in the text before.

This sentence has been removed.
